# Effect of sunflower seed oil emollient therapy on newborn infant survival in Uttar Pradesh, India: A community-based, cluster randomized, open-label controlled trial

**Aarti Kumar**[1], **Shambhavi Mishra**[1,2], **Shambhavi Singh**[1], **Sana Ashraf**[1], **Peiyi Kan**[3], **Amit Kumar Ghosh**[4], **Alok Kumar**[5], **Raghav Krishna**[1], **David K. Stevenson**[3], **Lu Tian**[6], **Peter M. Elias**[7], **Gary L. Darmstadt**[3], **Vishwajeet Kumar**[1]*, for the Shivgarh Emollient Research Group

**1** Community Empowerment Lab, Lucknow, Uttar Pradesh, India, **2** Department of Statistics, Lucknow University, Lucknow, Uttar Pradesh, India, **3** Prematurity Research Center, Division of Neonatal and Developmental Medicine, Department of Pediatrics, Stanford University School of Medicine, Stanford, California, United States of America, **4** Government of India, India, **5** Government of Uttar Pradesh, India, **6** Department of Health Research and Policy, Stanford University School of Medicine, Stanford, California, United States of America, **7** Department of Dermatology, University of California, San Francisco, California, United States of America

* vkumar@celworld.org

**Data Availability Statement:** The study data and data dictionary are available at https://doi.org/10.7910/DVN/TGNC9H, Harvard Dataverse, V1.

## Abstract

### Background

Hospitalized preterm infants with compromised skin barrier function treated topically with sunflower seed oil (SSO) have shown reductions in sepsis and neonatal mortality rate (NMR). Mustard oil and products commonly used in high-mortality settings may possibly harm skin barrier integrity and enhance risk of infection and mortality in newborn infants. We hypothesized that SSO therapy may reduce NMR in such settings.

### Methods and findings

This was a population-based, cluster randomized, controlled trial in 276 clusters in rural Uttar Pradesh, India. All newborn infants identified through population-based surveillance in the study clusters within 7 days of delivery were enrolled from November 2014 to October 2016. Exclusive, 3 times daily, gentle applications of 10 ml of SSO to newborn infants by families throughout the neonatal period were recommended in intervention clusters ($n = 138$ clusters); infants in comparison clusters ($n = 138$ clusters) received usual care, such as massage practice typically with mustard oil. Primary analysis was by intention-to-treat with NMR and post-24-hour NMR as the primary outcomes. Secondary analysis included per-protocol analysis and subgroup analyses for NMR. Regression analysis was adjusted for caste, first-visit weight, delivery attendant, gravidity, maternal age, maternal education, sex of the infant, and multiple births. We enrolled 13,478 (52.2% male, mean weight: 2,575.0 grams ± standard deviation [SD] 521.0) and 13,109 (52.0% male, mean weight: 2,607.0 grams ± SD 509.0) newborn infants in the intervention and comparison clusters,

**Funding:** The trial was funded by the World Health Organization (WHO, https://www.who.int). VK was the primary recipient of the grant from the WHO for the submitted work under Project ID HQMCA1611093, Award 61748, which was also used to support AK, SM, SA, SS and RK. Other authors AKG, AIK, DS, LT, PE and GLD received no specific funding for this work. The WHO provided inputs into the study design, oversight into the conduct of the trial, coordinated the ethical review process at WHO, convened the TAG and DSMB, and coordinated the reporting of SAEs to the DSMB. They had no role in data collection and analysis, decision to publish, or preparation of the manuscript.

**Competing interests:** I have read the journal's policy and the authors of this manuscript have the following competing interests: PE is a co-developer of EpiCeram, a treatment for Atopic Dermatitis, currently licensed to Primus Pharmaceuticals, Scottsdale, AZ (with whom he is engaged as consultant) by the University of California.

**Abbreviations:** aHR, adjusted hazard ratio; aOR, adjusted odds ratio; CI, confidence interval; CONSORT, Consolidated Standards of Reporting Trials; DSMB, Data and Safety Monitoring Board; IQR, interquartile range; LBW, low birth weight; NMR, neonatal mortality rate; SAE, severe adverse event; SD, standard deviation; SSO, sunflower seed oil; TAG, technical advisory group; VLBW, very low birth weight; WHO, World Health Organization.

respectively. We found no overall difference in NMR in the intervention versus the comparison clusters [adjusted odds ratio (aOR) 0.96, 95% confidence interval (CI) 0.84 to 1.11, $p = 0.61$]. Acceptance of SSO in the intervention arm was high at 89.3%, but adherence to exclusive applications of SSO was 30.4%. Per-protocol analysis showed a significant 58% (95% CI 42% to 69%, $p < 0.01$) reduction in mortality among infants in the intervention group who were treated exclusively with SSO as intended versus infants in the comparison group who received exclusive applications of mustard oil. A significant 52% (95% CI 12% to 74%, $p = 0.02$) reduction in NMR was observed in the subgroup of infants weighing $\leq$1,500 g ($n = 589$); there were no statistically significant differences in other prespecified subgroup comparisons by low birth weight (LBW), birthplace, and wealth. No severe adverse events (SAEs) were attributable to the intervention. The study was limited by inability to mask allocation to study workers or participants and by measurement of emollient use based on caregiver responses and not actual observation.

## Conclusions

In this trial, we observed that promotion of SSO therapy universally for all newborn infants was not effective in reducing NMR. However, this result may not necessarily establish equivalence between SSO and mustard oil massage in light of our secondary findings. Mortality reduction in the subgroup of infants $\leq$1,500 g was consistent with previous hospital-based efficacy studies, potentially extending the applicability of emollient therapy in very low-birth-weight (VLBW) infants along the facility–community continuum. Further research is recommended to develop and evaluate therapeutic regimens and continuum of care delivery strategies for emollient therapy for newborn infants at highest risk of compromised skin barrier function.

## Trial registration

ISRCTN Registry ISRCTN38965585 and Clinical Trials Registry—India (CTRI/2014/12/005282) with WHO UTN # U1111-1158-4665.

Author summary

### Why was this study done?

- Preterm infants have an immature skin barrier, making them susceptible to systemic infections. In animal models, some emollients such as sunflower seed oil (SSO) were found to enhance skin barrier integrity, and some such as mustard oil were found to impair skin barrier integrity.

- Hospital-based studies with emollient therapy in preterm infants showed a 50% reduction in bloodstream-proven infections and 27% reduction in neonatal mortality rate (NMR).

- We hypothesized that in community settings in South Asia with near-universal prevalence of vigorous massage with mustard oil, all newborn infants, regardless of maturity,

may be susceptible to impaired skin barrier integrity, and, therefore, may benefit from SSO therapy leading to NMR reduction.

### What did the researchers do and find?

- We conducted a community-based cluster randomized controlled trial to evaluate the impact of SSO therapy as compared to usual massage practices in newborn infants.

- Primary intention-to-treat analysis found no difference in NMR between the intervention and control arms.

- Secondary analysis found 52% lower NMR in the subgroup of very low-birth-weight (VLBW) infants (≤1,500 g) and 58% lower NMR in all newborn infants treated exclusively with SSO in intervention clusters compared to those who exclusively received mustard oil massage in the comparison clusters.

### What do these findings mean?

- Our primary findings do not support recommendations for universal promotion of SSO therapy for NMR reduction.

- Secondary findings in VLBW infants appear consistent with previous hospital-based studies and suggest possible benefit of extending emollient therapy in this subgroup along the facility–community continuum, which needs to be evaluated through an appropriately designed study.

### Introduction

Experimental trials of topical emollient therapy—primarily with sunflower seed oil (SSO)—in hospitalized very preterm infants <33 weeks gestational age have demonstrated a 50% reduction in bloodstream infections and 27% reduction in neonatal mortality rate (NMR) [1–5]. The skin barrier is developmentally compromised and easily injured in preterm infants—increasingly so with advancing degree of prematurity—posing a risk for accelerated water and heat loss, entry of pathogens, and mortality [6–10]. Natural oils differ in their barrier promoting properties based on their fatty acid composition [11,12]. In studies using mouse models of human infant skin, high-linoleate SSO was found to enhance skin barrier integrity and repair, whereas certain products and oils, such as mustard oil, which are routinely applied to newborn infants in high-NMR regions in South Asia and sub-Saharan Africa, showed harmful effects [12,13]. We assumed that widespread newborn skin care practices in these settings, such as forceful massage and removal of the vernix after birth, may place all newborn infants, regardless of gestational age or birth weight, at elevated risk of morbidity and mortality by injuring natural protective barriers in a context of high exposure to environmental pathogens [14,15]. We therefore hypothesized that SSO therapy recommended for all newborn infants may improve neonatal survival in high-NMR community settings.

We examined the impact of promoting SSO therapy compared to existing skin care practices—which almost uniformly include mustard oil massage—on NMR and post-24-hour NMR in a high-NMR community setting in the state of Uttar Pradesh, India. Unlike previous hospital-based studies in preterm infants, this trial evaluated emollient therapy application on all newborn infants by families (rather than health workers) in a community setting across the facility–community continuum, with usual skin care practices (rather than no emollient application) as the comparison and with a primary outcome of mortality. The study has high potential generalizability as socioeconomic characteristics and cultural practices in this setting are similar to other high-mortality regions in South Asia that together account for about 40% of the global burden of neonatal deaths.

## Methods

This study is reported as per the Consolidated Standards of Reporting Trials (CONSORT) guideline (S1 CONSORT Checklist).

### Study design

The study was designed as a 2-arm cluster randomized controlled efficacy trial [16]. The study area was spread across 276 contiguous villages in a rural population of approximately 818,000 in the demographic surveillance area of the Community Empowerment Lab in Uttar Pradesh (Fig 1). Each village was considered as a unique cluster. As the intervention involved modifying preexisting, near-universal practices, randomization was done at the cluster level to minimize contamination of intervention into comparison clusters. NMR and post-24-hour NMR were independent primary outcomes of the trial. We included post-24-hour NMR as a primary outcome to potentially improve the power of the study, assuming that the intervention was unlikely to impact mortality within the first 24 hours.

The sample size was estimated to enable detection in intention-to-treat analysis of at least 15% reduction in NMR and 20% reduction in post-24-hour NMR at 5% level of significance with 90% power [17]. A mean NMR of 45 per 1,000 was assumed [18], with a range of ±15% across control clusters, yielding a between-cluster coefficient of variation of 0.10 [17]. Calculations assumed 162 births per cluster (mean population: approximately 3,000 with crude birth rate of 27 per 1,000 over an enrolment period of 24 months), expected loss to follow-up of 8%, yielding a sample size of 138 clusters per study arm (or 20,536 newborn infants in each arm) with NMR as the primary outcome. The sample size calculated for post-24-hour NMR was lower at 132 clusters [16], and, therefore, $n = 276$ clusters was taken as the final sample size.

Randomization was conducted at the World Health Organization (WHO), allocating all 276 clusters equally to the 2 study arms while applying restriction criteria that allowed for a maximum overall difference of 50 in mean number of households, 2% in number of Muslim households, 2% in number of scheduled caste households, and 2 per 1,000 in baseline NMR between intervention and control clusters. Of 10,000 simple randomization schemes generated, 1,944 met the restriction criteria, of which one was randomly chosen, and its sequences randomly allocated to intervention and control. Allocated sequences were emailed to the principal investigator. All infants identified in study clusters within 7 days of delivery and whose family planned to stay within the study area were enrolled into the study cluster where they were first identified, irrespective of subsequent migration. There were no prespecified exclusion criteria. Due to the nature of the intervention, it was impossible to mask allocation; however, the intervention and evaluation teams were independent with separate lines of management and communication in order to minimize potential for bias. Data management protocols masked the cluster allocation from monitoring and analysis teams, which ensured

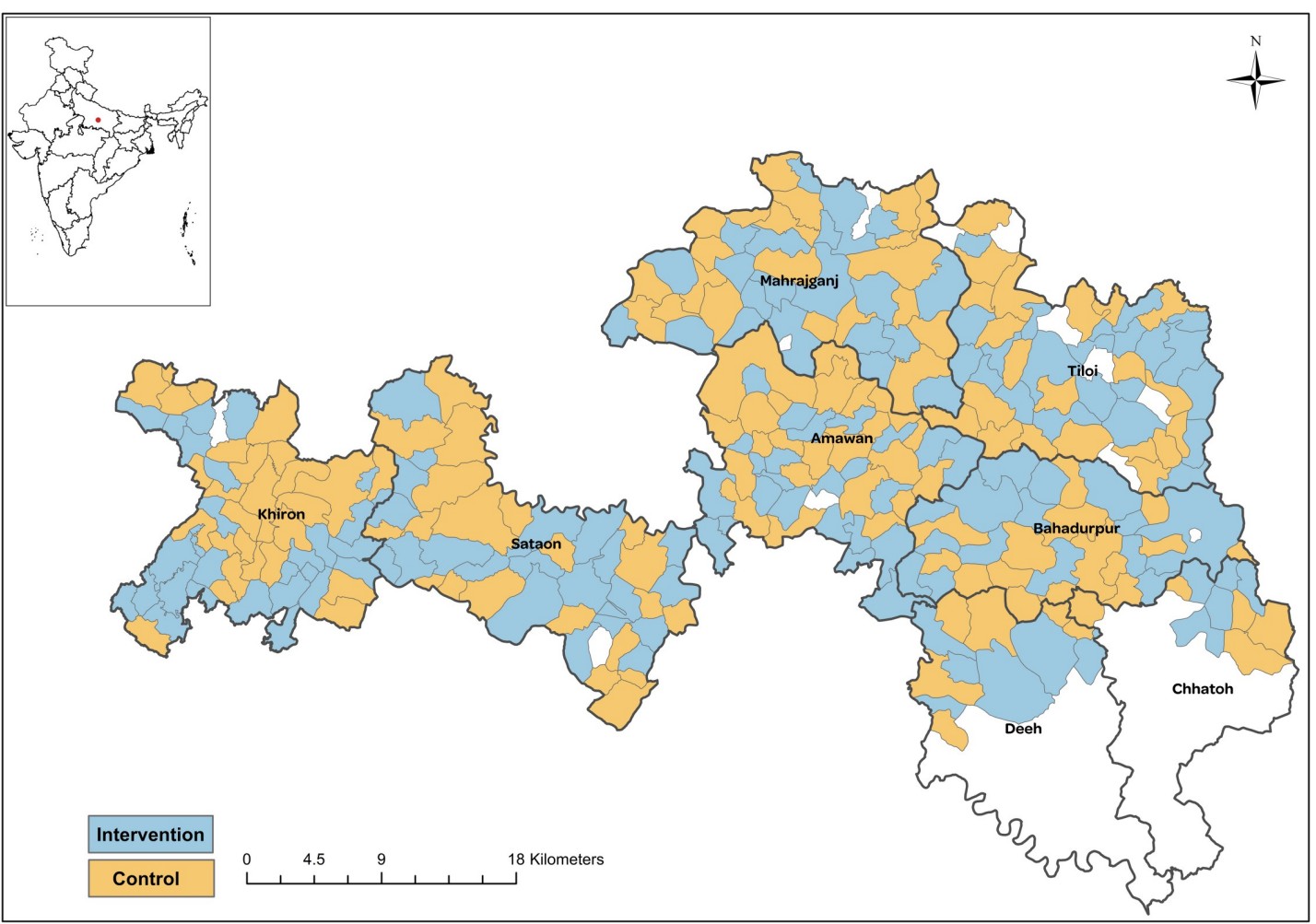

**Fig 1. Trial location in Raebareli district, Uttar Pradesh, India, showing the randomized allocation of intervention and comparison clusters across study blocks.** The base map was created at the Community Empowerment Lab using publicly available data.

that during the execution of the study, data were never accessed or reported separately for the 2 study groups.

Prospective 2-monthly, door-to-door pregnancy surveillance and delivery notification systems were established and augmented by a social network of key informants. Information on pregnancy and delivery notification was processed centrally at a data management center and relayed by a call center to both intervention and evaluation teams to trigger their respective protocols.

## Intervention

Formative research found that vigorous massage of newborn infants, typically with mustard oil, sometimes augmented with herbs and homemade cleansing scrub pastes, was a community norm in this region. There was a very strong intrinsic belief in the goodness of mustard oil and its ability to strengthen the bones of infants. Newborn infants typically receive oil massage 2 to 5 times daily [14]. Oil application typically begins early, with the first application used to facilitate removal of the vernix and cleansing immediately after birth. In the control clusters, no effort was made to change these usual practices.

The recommended dosage of SSO was based on the protocol followed in previous hospital-based studies of SSO therapy that showed NMR impact in preterm infants. In the intervention clusters, families were provided natural, cold-pressed, high-linoleate SSO in light and heat-protected, tamper-proof, sterilized containers designed to maintain product integrity. The container was branded following a human-centered design process to enhance its desirability and adoption and enable detection of contamination of use of intervention oil in control clusters. We recommended gentle applications of 10 ml of SSO with washed hands 3 times daily throughout the neonatal period. Each batch of SSO was quality tested in an accredited laboratory (SGS India, Gurgaon, India) for its linoleic acid content (>60%)—a key ingredient—and absence of harmful chemicals.

Our behavior change strategy emphasized early and exclusive use of SSO and nonuse of any other products or emollients on the skin for all newborn infants, regardless of birth weight. We did not want to attack the community's belief in the goodness of mustard oil, and, therefore, our behavior change strategy simply promoted SSO as an oil that was suited to the fragile skin of the newborn, which was like the petal of a flower, and asked them to defer application of mustard oil and homemade scrub "bukwa", until after the newborn period.

Community-based female workers with 12 or more years of formal education and no prior exposure to any health intervention–based work were chosen as intervention workers. Their role involved following up pregnant women and newborn infants, distributing the SSO bottles as per schedule, promoting exclusive use of SSO, and counseling families on the method of application. Recognizing that there were no corresponding visits comparable to intervention worker visits in the control arm, intervention workers received no training on newborn care and were expressly instructed to refrain from counseling families on other aspects of newborn care besides the intervention itself. Intervention workers promoted and distributed, but did not apply, SSO, which was left to families and traditional masseuses. They received 3 days of classroom training on SSO application but no training on any additional aspects of newborn care and were instructed to refrain from counseling families on any other care practices besides the intervention itself. Each worker was responsible for 2 intervention clusters and visited pregnant women in their 25th week of pregnancy and on the first and seventh days after birth. The antenatal visit was timed to ensure coverage of very premature infants and involved written informed consent, counseling on SSO application, and provision of a 100-ml bottle of SSO to facilitate initiation of the intervention immediately after delivery in the hospital or at home. The postnatal day 1 visit involved observation of SSO application by the family, counseling, and provision of a 200-ml bottle of SSO. Three additional 200-ml bottles of SSO were provided on postnatal day 7 to last till the remainder of the neonatal period. Traditional masseuses in the community were typically engaged by families to perform once daily massage of newborn infants, while the family performed all other applications. In the intervention clusters ($n$ = 1,189) we provided traditional masseuses a 4-hour group orientation and training on SSO application in order to enlist their voluntary support in promoting adherence to the intervention by families. Additionally, monthly community meetings involving pregnant women and their families, mothers who were championing use of SSO, and traditional masseuses were conducted to reinforce the early and exclusive application of SSO through gentle massage. Control clusters did not receive any visits, oil, promotion, or engagement of traditional masseuses. However, there were exactly the same number of intervention and evaluation worker visits scheduled on the same days both antenatally and postnatally. This was done to minimize the possibility of bias due to a "visitation effect," as the evaluation worker visits were exactly the same in both the arms.

## Evaluation

Female evaluation workers had 12 or more years of education and were provided 7 days of classroom and 3 days of field-based training on data collection procedures including use of tablet devices and weighing of newborn infants. Each data collector covered approximately 4 clusters, which could be a combination of intervention and control. They gathered data on socioeconomic status and pregnancy history during the antenatal visit in the 25th week of pregnancy. They enrolled newborn infants and recorded delivery details, including date and time of delivery, during the first postnatal visit scheduled on day 1. Data on newborn survival status, date and time of death if applicable, and oil application focusing on the time between visits were collected during postnatal visitations on days 1 (recall period from birth to the first postnatal visit), 7 (recall period from the first to the second visit), and 29 after delivery (recall period from the second to the third visit). We collected data on skin products used and frequency of massage but did not have reliable means of assessing other practice outcomes such as gentleness of massage technique and amount of oil applied. Newborn weighing was done during the day 1 visit as a proxy for birth weight using hanging sling digital weighing scales AMW-SR-20 (precision ± 10 g, American Weigh Scales, Cumming, Georgia, USA) under strict protocols to ensure accuracy, safety, and hygiene. Age at death was calculated as the difference between date and time of death and date and time of delivery. Accordingly, neonatal death was classified as death that occurred before completion of 28 days of life, and post-24-hour neonatal death was classified as death that occurred after 24 hours but before completion of 28 days of life.

Severe adverse events (SAEs), including neonatal deaths and stillbirths, hospitalization, and skin rash/skin reactions/skin infections, were either reported by families or identified during the scheduled data collection visits. A specialized team of data collectors was constituted for confirmation of all reported SAEs using a standardized questionnaire and observation protocol. SAEs were reported to WHO upon confirmation within 3 days of identification. WHO monitored and reported SAEs to the Data and Safety Monitoring Board (DSMB).

## Data management and quality assurance

Data collection was done on a proprietary software on tablet devices and was synchronized from the cloud to a local on-site MySQL Database on a daily basis. Forms included in-built checks for logical inconsistencies, skips, missing values, and range limits. Follow-up visitations by data collectors were centrally scheduled and dispatched to their tablet devices and monitored through a call center. On-field quality checks, consisting of spot checks (5%) to observe the data collection process, back checks (5%) to readminister a random list of questions for verification, and verification of all stillbirths and neonatal deaths, were conducted by a team of supervisors who were each responsible for 10 workers. A team of 2 data analysts blinded to study arms reviewed the data quality on a weekly basis and provided specific feedback on data completeness, distributions of observed values, and inconsistencies.

## Statistical analysis

**Primary analysis (intention-to-treat).** Primary analysis was done based on principles of intention-to-treat in SAS v9.0 (SAS Institute, Cary, North Carolina, USA) and verified independently in STATA v13.0 (StataCorp, College Station, Texas, USA). Newborn infants were analyzed as part of the cluster where they were first identified, irrespective of cross-migration. For analysis of neonatal mortality, our predetermined analysis plan (S1 Analysis Plan) had specified a comparison of cluster-level NMR across groups using a 2-sample $t$ test. However, due to wide variation in cluster sizes, this method was deemed inappropriate, and we adopted a more statistically efficient method of individual-level analysis adjusted for within-cluster

correlation using a random effects logistic regression model [17] recommended by statisticians blinded to the data. Regression analysis was adjusted for caste, first-visit weight (as a proxy for birth weight), delivery attendant, gravidity, maternal age, maternal education, sex of the infant, and multiple births. This method was applied for the primary outcomes, NMR and post-24-hour NMR. We also conducted a sensitivity analysis to estimate the treatment effect on survival time using Kaplan–Meier curves (taking day 0 as the day of birth and day 27 as the 28th completed day of the neonatal period) and performed proportional hazards regression adjusted for clustering and the covariates listed above.

**Subgroup analysis.** Subgroup analyses for NMR were prespecified based on prematurity, low birth weight (LBW), birthplace, wealth, and emollient use. Methods for intention-to-treat analyses were replicated in subgroups of infants based on (1) prematurity: first-visit weight ≤1,500 g (i.e., very low birth weight, VLBW) [19] as a proxy for very preterm in the absence of ultrasound reports and reliable data from recall of last menstrual period, i.e., in the absence of reliable measures of gestational age; (2) LBW: first-visit weight ≤2,500 g; (3) birthplace: home and health facility births; (4) wealth: tertiles (rather than quintiles as originally planned since household wealth was clustered and model convergence could not be achieved without combining the 3 lowest quintiles into 1 category) generated using principal component analysis applied to baseline asset and household characteristic data; and (5) emollient use: analyzed as per-protocol analysis (below). We performed survival analysis on subgroups that were significantly associated with NMR.

**Per-protocol analysis.** To assess the efficacy of SSO therapy, we conducted per-protocol analysis using data on oil application. The follow-up period was from birth to death or 28 days of age. Complete adherence to SSO was defined as using SSO to initially cleanse the newborn infant within minutes of birth, using SSO for the first oil application within 6 hours of delivery, and subsequently applying SSO exclusively 3 or more times a day during the entire follow-up period. An analogous definition was used to identify newborn infants who were treated exclusively with mustard oil. Infants treated with additional regimens (e.g., more than 1 oil) were not included due to the biased association of increased likelihood of alternative regimens with increased survival time. We compared participants who had been randomized to the intervention group and who strictly adhered to the exclusive use of SSO ("exclusive SSO") to those in the comparison group who used mustard oil exclusively ("exclusive mustard oil"), analyzing data for neonatal mortality using the same regression methods as for the intention-to-treat analysis. While our intervention also recommended gentle massage against the traditional technique of vigorous massage, we did not have any reliable indicator to measure whether this was indeed practiced—hence, the per-protocol analysis only focuses on the oil used and not the technique. We conducted an exploratory sensitivity analysis comparing infants who strictly adhered to the exclusive use of SSO to those who used mustard oil exclusively, without regard to random allocation to treatment group.

## Ethics approval

The study received ethical clearance from the Ethics Review Committee at WHO in Geneva, Switzerland, which was the study sponsor, and the Institutional Ethics Committee at the Community Empowerment Lab in Uttar Pradesh. Community-level consent was obtained from community leaders in each of the study clusters prior to randomization, and written informed consent was obtained from the parents/guardians of all infants prior to enrolment.

## Study oversight

A technical advisory group (TAG) and a DSMB were convened by WHO. WHO and the TAG provided inputs into the study design. Study oversight involved weekly calls, quarterly visits by WHO oversight team, and annual visits by the TAG and DSMB teams.

The DSMB conducted a planned, masked, unadjusted analysis of 50% of the originally estimated sample size (based on follow-up until June 2016) during its midterm review meeting in September 2016 and recommended that the investigators suspend further recruitment by 15 October 2016, complete follow-up of participants already enrolled in the trial—which was achieved on November 22—and send the complete data to the DSMB for review and a decision on further action. The DSMB reviewed an unmasked analysis of this dataset conducted at WHO consisting of approximately 70% of the originally estimated sample size in its meeting held January 16, 2017. The DSMB concluded that (1) there were no safety concerns and study stopping rules for futility were not met ("... the study should be continued until the originally proposed sample size was reached, unless futility was accompanied by indications of harm in which case the rules for safety would apply"); and (2) the "confidence intervals indicated that a beneficial effect of more than 11% on post-24 hour neonatal mortality and more than 13% on overall neonatal mortality was extremely unlikely," and, therefore, the null hypotheses of the study was unlikely to be rejected. The DSMB left the final decision regarding continuation of further enrolment into the study to the sponsor and investigators and recommended: "There is no need to restart enrolment as the study questions have already been answered." Further, the suspension of enrolment and intervention activities for over 3 months made it unfeasible to restart the study at such a large geographical scale. Consequently, the sponsor and investigators mutually agreed to officially close the study—with no safety concerns and without reaching predefined rules for futility—on February 3, 2017, prior to completion of sample size requirements.

## Results

### Study population

Study clusters ($n$ = 276) were equally allocated to intervention and comparison arms, and all clusters were enrolled (Figs 1 and 2). From November 2014 to October 2016, pregnancy surveillance identified 15,720 and 15,451 women in the intervention and comparison arms, respectively, of whom 13,779 (87.7%) and 13,463 (87.1%) women met the study inclusion criteria. All live-born infants (13,478 intervention, 13,109 comparison) were enrolled and followed up through the neonatal period in both the arms. No clusters or participants were lost to follow-up.

Baseline characteristics were comparable across the 2 randomized study groups (Table 1). Mothers had a mean age of 25.4 years, 86.7% were Hindu with 35.9% from scheduled castes, 33.3% were illiterate, 90.0% practiced open defecation, and 39.1% were primiparous. Most (84.8%) women delivered in a healthcare facility with a skilled birth attendant (83.1%). The male-to-female ratio among livebirths was 1.09:1. At the first postnatal (enrolment) visit that occurred on day 2 (interquartile range [IQR]: days 1 to 3) for both intervention and comparison arms (Table 1), 589 infants (2.2%) weighed ≤1,500 g. SSO composition was tested for each of the 8 batches of oil procured to ensure that they met food safety standards; there was minimal variation in linoleic acid content (61.5% ± 1.6%). We also report baseline characteristics separately for secondary per-protocol analysis, which were similar for the participants in the intervention arm who adhered to exclusive use of SSO and those in the comparison arm who practiced exclusive use of mustard oil, with minor differences such as caste composition and delivery attendant that were adjusted for in our analyses (Table 1).

### Massage practices

**Use of oil.** In the intervention arm, 89.3% reported any application of SSO, and 61.3% reported any application of mustard oil during the newborn period. Corresponding values for

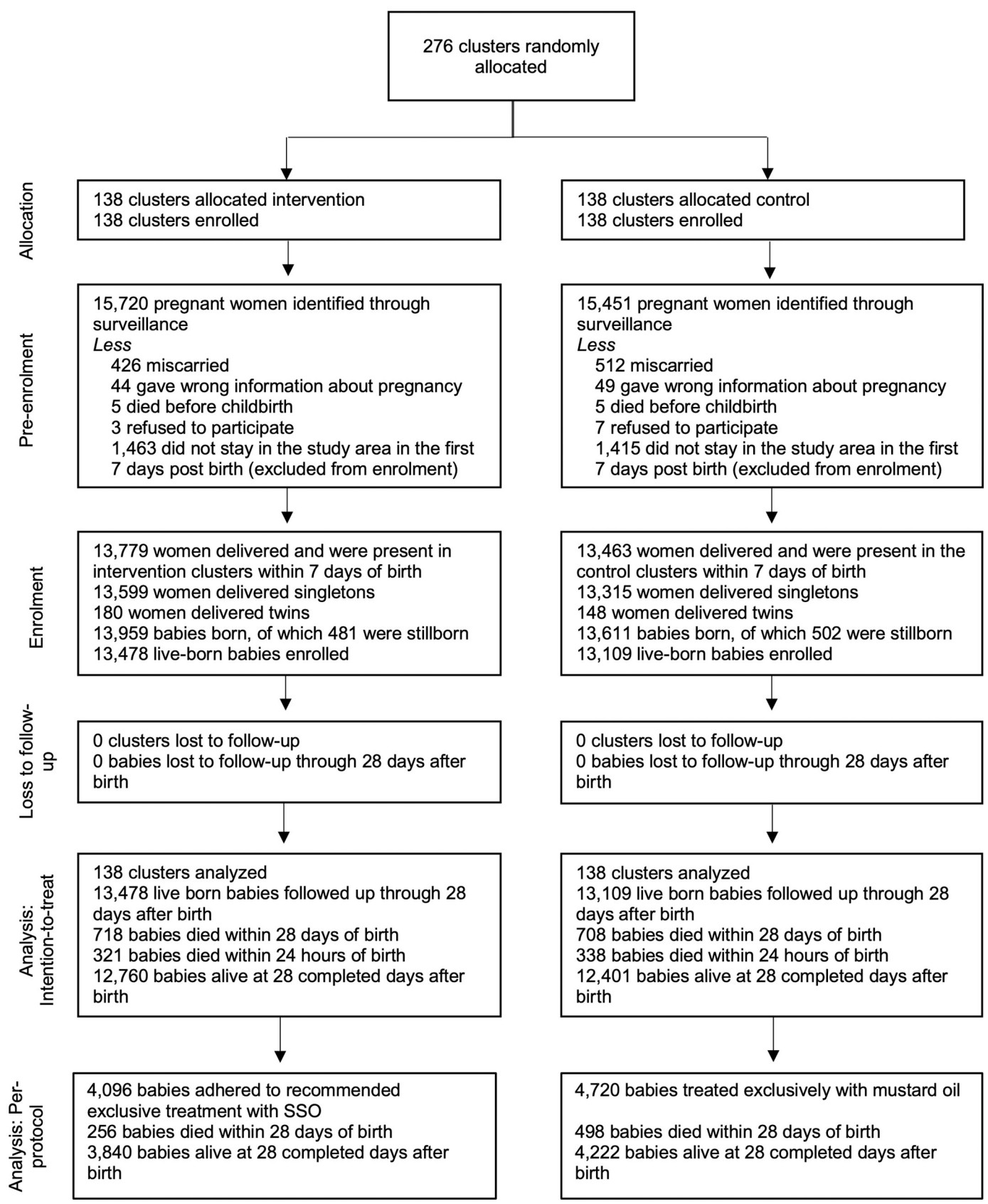

**Fig 2. Participant flow diagram.** SSO, sunflower seed oil.

**Table 1.** Baseline characteristics of the study population randomized to intervention and comparison groups for intention-to-treat analysis and included in per-protocol analysis.

| Characteristic | Intention-to-treat | | Per protocol | |
|---|---|---|---|---|
| | Comparison [N (%)] | Intervention [N (%)] | Comparison: exclusive mustard oil [N (%)] | Intervention: exclusive SSO [N (%)] |
| **Households per cluster [median N (range)]** | 458 (208–2,286) | 477 (210–3,061) | | |
| **Cluster size [median N IQR]** | 80 (60–113) | 82.5 (59–119) | | |
| **Total live births [N (%)]** | 13,109 (49.3) | 13,478 (50.7) | 4,720 (53.5%) | 4,096 (46.5%) |
| **Male [N (%)]** | 6,820 (52.0) | 7,042 (52.2) | 2,431 (51.5) | 2,114 (51.6) |
| **Religion [N (%)]** | | | | |
| Hindu | 11,346 (86.6) | 11,675 (86.6) | 4,185 (88.7) | 3,502 (85.5) |
| Muslim | 1,743 (13.3) | 1,787 (13.3) | 532 (11.3) | 588 (14.4) |
| Other | 14 (0.1) | 13 (0.1) | 3 (0.1) | 6 (0.1) |
| **Maternal age (mean years ± SD)** | 25.3 ± 3.7 | 25.4 ± 3.7 | 25.4 ± 3.8 | 25.5 ± 3.7 |
| **Caste [N (%)]** | | | | |
| General | 2,038 (15.6) | 2,001 (14.9) | 676 (14.3) | 560 (13.7) |
| Other backward caste | 6,372 (48.6) | 6,622 (49.1) | 2,165 (45.9) | 2,003 (48.9) |
| Scheduled caste/scheduled tribe | 4,693 (35.8) | 4,852 (36.0) | 1,879 (39.8) | 1,533 (37.4) |
| **Maternal education [N (%)]** | | | | |
| Illiterate | 4,324 (33.0) | 4,531 (33.6) | 1,536 (32.5) | 1,416 (34.6) |
| Primary completed | 2,913 (22.2) | 3,215 (23.9) | 1,050 (22.2) | 949 (23.2) |
| Tenth grade completed | 3,843 (29.3) | 3,720 (27.6) | 1,451 (30.7) | 1,168 (28.5) |
| Secondary and above completed | 2,023 (15.4) | 2,009 (14.9) | 683 (14.5) | 563 (13.7) |
| **Delivery place [N (%)]** | | | | |
| Government facilities | 10,092 (77.0) | 10,476 (77.7) | 3,586 (76.0) | 3,360 (82.0) |
| Private facilities | 946 (7.2) | 1,021 (7.6) | 292 (6.2) | 198 (4.8) |
| On the way to a facility from home | 40 (0.3) | 30 (0.2) | 19 (0.4) | 7 (0.2) |
| Home | 2,031 (15.5) | 1,949 (14.5) | 823 (17.4) | 531 (13.0) |
| **Delivery attendant [N (%)]** | | | | |
| Government doctor | 3,285 (25.1) | 4,024 (29.9) | 1,176 (24.9) | 1,045 (25.5) |
| Private doctor | 774 (5.9) | 907 (6.7) | 244 (5.2) | 183 (4.5) |
| Auxiliary nurse midwife/staff nurse | 6,710 (51.2) | 6,397 (47.5) | 2,374 (50.3) | 2,273 (55.5) |
| Others | 2,340 (17.9) | 2,148 (15.9) | 926 (19.6) | 595 (14.5) |
| **Delivery type [N (%)]** | | | | |
| Normal | 12,136 (92.6) | 12,407 (92.1) | 4,396 (93.1) | 3,891 (95.0) |
| Assisted (forceps) | 169 (1.3) | 181 (1.3) | 60 (1.3) | 27 (0.7) |
| Normal with episiotomy | 501 (3.8) | 563 (4.2) | 151 (3.2) | 108 (2.6) |
| Cesarean | 303 (2.3) | 325 (2.4) | 113 (2.4) | 70 (1.7) |
| **Gravidity [N (%)]** | | | | |
| 1 | 5,105 (39.0) | 5,282 (39.2) | 1,841 (39.0) | 1,594 (38.9) |
| 2 | 3,254 (24.8) | 3,177 (23.6) | 1,225 (26.0) | 979 (23.9) |
| 3 | 2,245 (17.1) | 2,293 (17.0) | 747 (15.8) | 681 (16.6) |
| 4+ | 2,499 (19.1) | 2,723 (20.2) | 907 (19.2) | 842 (20.6) |
| **Toilet type [N (%)]** | | | | |
| Open defecation | 11,718 (89.4) | 12,203 (90.6) | 4,223 (89.5) | 3,767 (92.0) |
| Flush toilet | 816 (6.2) | 750 (5.6) | 288 (6.1) | 202 (4.9) |
| Pit latrine | 304 (2.3) | 365 (2.7) | 144 (3.1) | 88 (2.1) |
| Public toilet | 265 (2.0) | 154 (1.1) | 65 (1.4) | 37 (0.9) |
| Other | 0 | 3 (0) | 0 | 2 (0) |

*(Continued)*

**Table 1.** (Continued)

| Characteristic | Intention-to-treat | | Per protocol | |
| --- | --- | --- | --- | --- |
| | Comparison [N (%)] | Intervention [N (%)] | Comparison: exclusive mustard oil [N (%)] | Intervention: exclusive SSO [N (%)] |
| Age at measurement of first-visit weight (median days ± IQR) | 2 (1–3) | 2 (1–3) | 2 (2–4) | 2 (1–3) |
| First-visit weight, (mean grams ± SD) | 2,607.0 ± 509.0 | 2,575.0 ± 521.0 | 2,571.0 ± 530.5 | 2,575.7 ± 493.0 |

IQR, interquartile range; SD, standard deviation; SSO, sunflower seed oil.

the control arm were 2.8% for any use of SSO and 98.9% for any use of mustard oil. In the intervention arm, 4,096 infants received SSO therapy exclusively, an adherence rate of 30.4%, and in the comparison arm, 4,720 infants (36.0%) received mustard oil exclusively, with limited seasonal variation. Among 311 intervention infants weighing ≤1,500 g at birth, 73 were treated exclusively with SSO for an adherence rate of 23.5%. In the comparison arm, 146 of 278 (52.5%) infants weighing ≤1,500 g were treated exclusively with mustard oil.

**Frequency of massage.** Based on respondent recall from the second evaluation visit, intervention infants were massaged (regardless of oil used) for a mean of 2.68 (95% CI 2.66 to 2.70) times/day [median 3 and IQR (2 to 3)], whereas control infants were massaged for a mean of 2.38 (95% CI 2.37 to 2.40) times/day [median 2 and IQR (2 to 3)]. Based on the third visit recall, intervention infants were massaged (regardless of oil used) for a mean of 2.66 (95% CI 2.65 to 2.68) times/day [median 3 and IQR (2 to 3)], whereas control infants were massaged for a mean of 2.37 (95% CI 2.36 to 2.39) times/day [median 2 and IQR (2 to 3)].

## Intention-to-treat

Primary intention-to-treat regression analysis did not find a significant difference in mortality among newborn infants in the intervention (NMR 53.7 ± 2.3 per 1,000 livebirths) versus the comparison group (NMR 56.2 ± 2.4 per 1,000 livebirths) [adjusted odds ratio (aOR) 0.96, 95% confidence interval (CI) 0.84 to 1.11, $p = 0.61$) (Table 2). Results were similar for male (aOR 0.94, 95% CI 0.79 to 1.13) and female (aOR 1.00, 95% CI 0.80 to 1.25) infants. Similarly, there was no difference in post-24-hour NMR (Table 2). Sensitivity analysis based on survival time also showed no difference in survival of newborn infants in the intervention versus the comparison group (Fig 3A).

Subgroup analyses showed a significant 52% (95% CI 12% to 74%, $p = 0.02$) reduction in mortality among the subgroup of VLBW infants (2.2% of the study population) in the intervention (NMR 299.3 ± 37.8 per 1,000 livebirths) versus the comparison (NMR 402.4 ± 39.6 per 1,000 livebirths) group (Table 2). Survival analysis in this subgroup yielded a significant 35% reduced hazard of death (aHR 0.65, 95% CI 0.43 to 0.97, $p = 0.04$) in VLBW infants (Fig 3B). This survival advantage began on the first day after birth and occurred primarily in the first week but continued to accrue throughout the neonatal period. There was no significant treatment effect among other prespecified subgroups of LBW infants ≤2,500 g (40.8% of the study population) (Fig 3C), those born in health facilities (85%) or at home (15.0%), or in any wealth tertile (Table 2).

## Per protocol

Per-protocol analysis found that intervention group infants treated exclusively with SSO had significantly reduced NMR (91.5 ± 13.4) versus infants in the comparison arm who exclusively received mustard oil (NMR 217.7 ± 21.0) by regression analysis (aOR 0.42, 95% CI 0.31 to

**Table 2. Differences in (A) the primary outcome of NMR between intervention and comparison clusters of infants by intention-to-treat analysis, (B) NMR in subgroups of infants in the intervention versus the comparison group by intention-to-treat analysis, and (C) NMR of infants in the intervention group treated exclusively with SSO compared to infants in the comparison group massaged exclusively with mustard oil (per-protocol analysis).**

| Comparison group | | | Intervention group | | | Odds ratio[*] | |
|---|---|---|---|---|---|---|---|
| Livebirths (N) | Neonatal deaths (N) | NMR[**] per 1,000 livebirths | Livebirths (N) | Neonatal deaths (N) | NMR[**] per 1,000 livebirths | Unadjusted | Adjusted[+] |
| **A. Primary analysis (based on principles of intention-to-treat)** | | | | | | | |
| *All infants* | | | | | | | |
| 13,109 | 708 | 56.2 ± 2.4 k[‡] = 0.50 | 13,478 | 718 | 53.7 ± 2.3 | 0.97 (95% CI 0.87–1.10), $p = 0.67$ | 0.96 (95% CI 0.84–1.11), $p = 0.61$ |
| *Infants alive after 24 hours (post-24-hour NMR)* | | | | | | | |
| 12,771 | 370 | 30.0 ± 1.8 k[‡] = 0.70 | 13,157 | 397 | 30.1 ± 1.8 | 1.04 (95% CI 0.89–1.21), $p = 0.65$ | 1.02 (95% CI 0.86–1.20), $p = 0.85$ |
| **B. Subgroup analysis (based on principles of intention-to-treat)** | | | | | | | |
| *Premature (by proxy, VLBW) infants ≤1,500 g* | | | | | | | |
| 278 | 95 | 402.4 ± 39.6 | 311 | 78 | 299.3 ± 37.8 | 0.60 (95% CI 0.34–1.04), $p = 0.07$ | 0.48 (95% CI 0.26–0.88), $p = 0.02$ |
| *LBW infants ≤2,500 g* | | | | | | | |
| 5,236 | 304 | 63.7 ± 4.5 | 5,622 | 327 | 62.1 ± 4.2 | 0.99 (95% CI 0.83–1.19), $p = 0.98$ | 0.94 (95% CI 0.77–1.14), $p = 0.52$ |
| *Home births* | | | | | | | |
| 2,031 | 122 | 73.1 ± 9.4 | 1,949 | 90 | 50.5 ± 6.4 | 0.73 (95% CI 0.54–0.99), $p = 0.05$ | 0.81 (95% CI 0.55–1.19), $p = 0.27$ |
| *Facility births* | | | | | | | |
| 11,038 | 580 | 53.7 ± 2.5 | 11,497 | 625 | 55.2 ± 2.8 | 1.03 (95% CI 0.91–1.17), $p = 0.65$ | 0.99 (95% CI 0.85–1.16), $p = 0.93$ |
| *Low-income group (lower wealth tertile)* | | | | | | | |
| 4,288 | 261 | 59.4 ± 4.5 | 4,654 | 280 | 63.5 ± 5.2 | 0.99 (95% CI 0.83–1.18), $p = 0.89$ | 0.99 (95% CI 0.86–1.16), $p = 0.97$ |
| *Middle-income group (middle wealth tertile)* | | | | | | | |
| 4,368 | 251 | 60.2 ± 4.9 | 4,376 | 230 | 53.1 ± 4.0 | 0.90 (95% CI 0.74–1.1.0), $p = 0.29$ | 0.84 (95% CI 0.69–1.02), $p = 0.08$ |
| *High-income group (upper wealth tertile)* | | | | | | | |
| 4,452 | 196 | 47.3 ± 4.0 | 4,447 | 208 | 46.4 ± 3.7 | 1.04 (95% CI 0.84–1.30), $p = 0.53$ | 1.04 (95% CI 0.84–1.30), $p = 0.72$ |
| **C. Per-protocol analysis (exclusive mustard oil versus exclusive SSO)** | | | | | | | |
| *All infants* | | | | | | | |
| 4,720 | 498 | 217.7 ± 21.0 k[‡] = 0.50 | 4,096 | 256 | 91.5 ± 13.4 | 0.38 (95% CI 0.28–0.51), $p < 0.01$ | 0.42 (95% CI 0.31–0.58), $p < 0.01$ |

[*] Random effects logistic regression model with control as reference category, adjusted for clustering.

[**] Neonatal deaths per 1,000 live births, cluster means ± standard error.

[‡] Between-cluster coefficient of variation.

[+] Adjusted for caste, first-visit weight, delivery attendant, gravidity, mother's age, mother's education, infant's sex, and multiple births.

CI, confidence interval; LBW, low birth weight; NMR, neonatal mortality rate; SSO, sunflower seed oil; VLBW, very low birth weight.

0.58, $p < 0.01$) (Table 2) and by survival analysis (aHR 0.47, 95% CI 0.35 to 0.62, $p < 0.01$) (Fig 4). Sex-disaggregated data showed slightly higher reduction in NMR in male (aOR 0.40, 95% CI 0.27 to 0.58, $p < 0.01$) compared to female (aOR 0.53, 95% CI 0.35 to 0.79, $p < 0.01$) infants by regression analysis. Survival analysis suggests an early benefit in intervention infants treated exclusively with SSO, primarily in the first week after birth but continuing to accrue throughout the neonatal period (Fig 4). Exploratory sensitivity analysis that included all infants who were treated with exclusive SSO ($n = 4102$) compared to infants who received exclusive

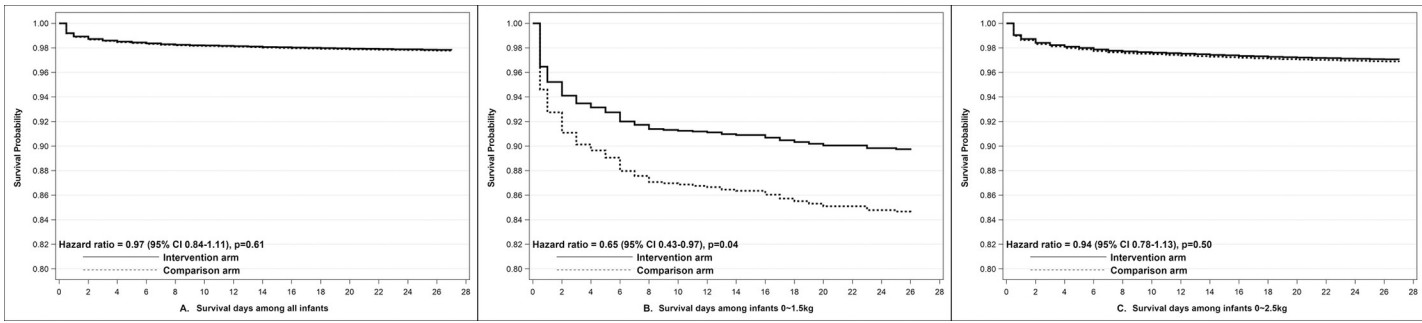

**Fig 3.** Kaplan–Meier survival curves showing treatment effect on survival time by intention-to-treat analysis for **(A)** all infants and subgroups of infants **(B)** ≤1,500 g and **(C)** ≤2,500 g. CI, confidence interval.

mustard oil (*n* = 5312), without regard to random group assignment, showed a 75% reduction in hazard of death (exclusive SSO aHR 0.25, 95% CI 0.20 to 0.31, *p* < 0.01, compared to exclusive mustard oil as the reference).

## Safety

No SAEs were attributable to the intervention. There were 512 versus 479 cases [unadjusted OR 1.04 (95% CI 0.92 to 1.18), *p* = 0.53] of skin rash/skin infection/skin reactions and 339 versus 320 cases [unadjusted OR 1.03 (95% CI 0.88 to 1.21), *p* = 0.70] of hospitalization reported in the intervention and control clusters, respectively.

## Discussion

The trial did not find any difference in NMR or post-24-hour NMR between newborn infants randomized to receive SSO therapy versus those randomized to receive usual care. Secondary analysis found 52% lower NMR in the subgroup of VLBW infants (≤1,500 g) and 58% lower NMR in all newborn infants treated exclusively with SSO in intervention clusters compared to those who exclusively received mustard oil massage in the comparison clusters. We did not find significant differences between intervention and control groups in the other prespecified subgroup comparisons by LBW, birthplace, and wealth.

This is the first community-based trial, to our knowledge, to report the effect of promotion of emollient therapy on newborn survival. Given prior evidence of significant benefit of SSO therapy in improving survival in VLBW infants in hospital settings [3,5], deleterious effects of mustard oil on skin barrier function observed in animal models [12], and our observation of seemingly harmful oil massage practices in communities in Uttar Pradesh [14,15], the above results merit closer examination and explanation. Besides the community setting, there were other important differences between this study and previous hospital-based studies on emollient therapy, including that (i) SSO therapy was recommended for all newborn infants regardless of birthplace and was not limited to very preterm/VLBW infants in hospitals; (ii) control infants received usual skin care practices such as mustard oil massage, whereas nothing was applied to control infants in previous studies; and (iii) SSO massage was provided by families and was not directly provided by intervention workers.

Several potential explanations must be considered. Low efficacy of SSO is unlikely, since oil quality and composition were similar to SSO used in previous hospital-based studies, and our SSO was subjected to stringent quality control procedures from production to distribution. Our assumption of deleterious effects of mustard oil massage on skin barrier function in human infants was based on animal models in the absence of prior data on human infants.

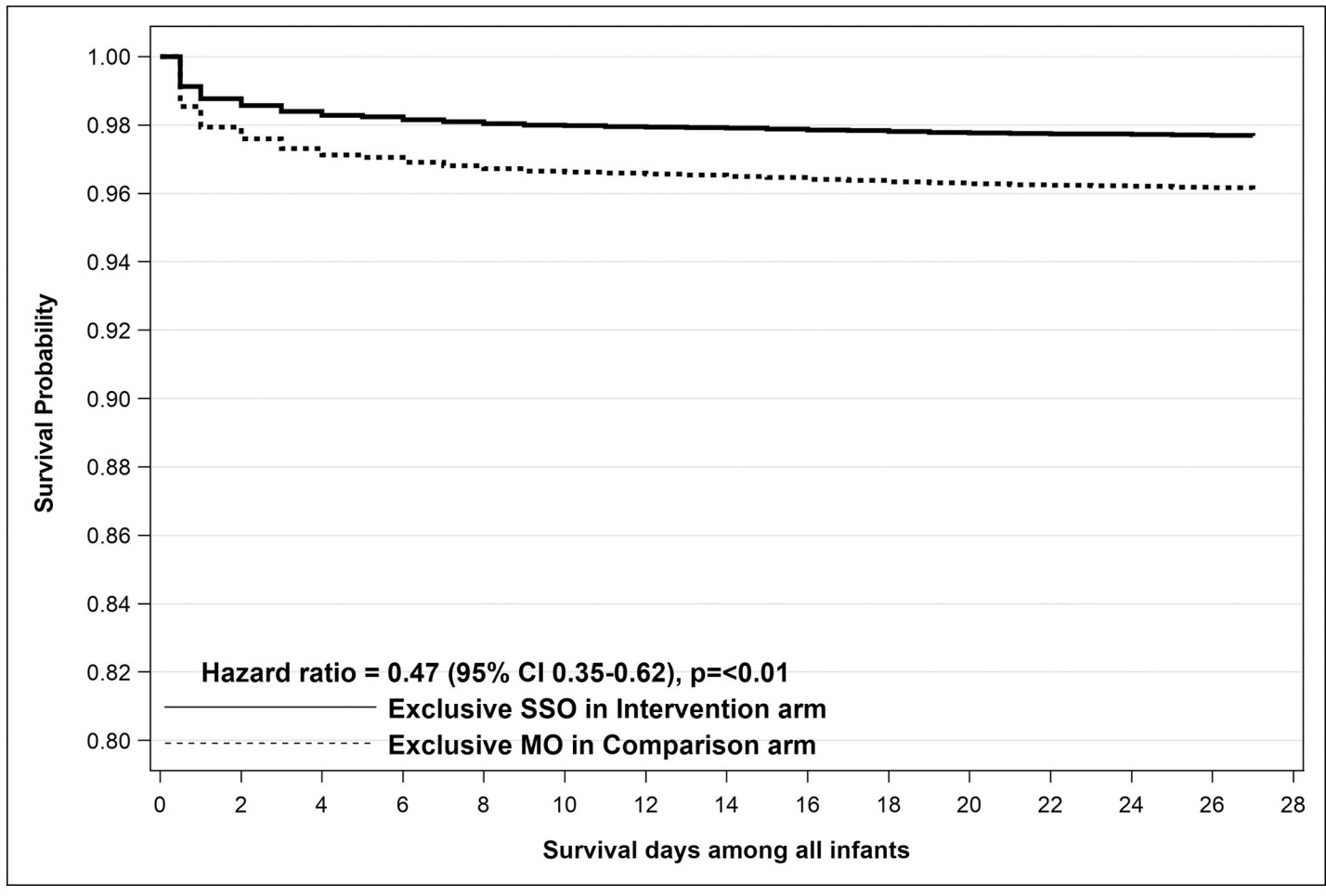

**Fig 4. Kaplan–Meier survival curve for per-protocol analysis comparing all infants randomized to the intervention group and treated exclusively with SSO versus infants randomized to the comparison group who exclusively received mustard oil.** CI, confidence interval; MO, mustard oil; SSO, sunflower seed oil.

Laboratory experiments in mouse models have demonstrated that mustard oil exerts toxicity to skin keratinocytes, whereas SSO substantially improves barrier function [12]. Recent data from mouse models indicate that products commonly applied to newborn infants in sub-Saharan Africa may also be detrimental to the skin barrier [13]. Results here suggest a possibility of protective clinical effects of SSO relative to mustard oil, especially in vulnerable VLBW infants. However, expected effects of SSO or mustard oil based on observations in controlled laboratory or hospital conditions may not necessarily bear out similarly in human infants in uncontrolled community settings. Recent studies from Nepal that for the first time assessed skin barrier integrity in human newborn infants in community settings did not find major differences between SSO and mustard oil groups [20]. Rate of reduction in skin pH during the first week of life was higher in infants massaged with SSO as compared to mustard oil, suggesting a faster acid mantle development and improved barrier normalization, but the improvements were modest [20]. No difference was observed in transepidermal water loss or skin scores and infants massaged with mustard oil fared slightly better in terms of a proxy measure for stratum corneum cohesion [20]. However, unlike previous hospital studies that involved gentle applications of SSO, infants in these studies from both the oil groups were massaged vigorously several times a day, and it is possible that massage technique may have influenced study outcomes

[20,21]. The study was limited by its inability to resolve the independent effects of the oils, massage practices, and environmental conditions [20,21].

Per-protocol analysis suggests that low adherence to exclusive SSO therapy may have contributed to the lack of population level impact. While acceptance of SSO was high in the intervention arm with 89.3% mothers reporting any application of SSO, adherence to exclusive application of SSO as recommended was only 30.4%. We compared the 30.4% of all infants in the intervention clusters who adhered to exclusive SSO treatment with the 36.0% of infants in the comparison clusters who were treated exclusively with mustard oil and found a significant 58% lower mortality in the former. Despite applying principles of behavior change management and human-centered design [14], behavior change of deeply rooted traditional practices was slower than anticipated [22]. Intervention worker visits to promote the intervention were limited to 3 communication-based interactions and relied on the family to initiate, adhere to, and sustain the therapy. We had decided on such a strategy due to its feasibility and scalability in community settings and to mitigate potential sources of bias due to a Hawthorne effect, which would be more likely to occur under intensive, highly controlled delivery of the intervention such as by ensuring that our own workers performed emollient application. The community's inherent belief in the goodness of mustard oil may have prevented them from giving up entirely on its use during the newborn period, while it may have been easier for them to accept a new oil for newborn massage that also appeared to be good. We did not consider it ethical to attack this preexisting belief in mustard oil in the absence of conclusive evidence of its inferiority in human newborn infants and also our inability to provide them with a continuous supply of SSO beyond the newborn period and beyond the duration of the study. One way of addressing these challenges with significant cost impact could have been to monitor intervention adherence and give time to achieve a certain level before initiating outcome evaluation. Adherence to exclusive SSO improved from 15.2% in June 2015 when implementation was rolled out in all clusters to 38.7% in September 2016 when enrolment was suspended, reflecting the need to allow sufficient time to shift deeply entrenched community practices in studies of this nature. Early truncation of the trial also limited the sample size and shortened the period available for achieving higher levels of adherence. The observed increase in adherence to exclusive SSO application over time suggests that there may have been potential to improve it further, and if this was achieved, overall population-level benefit could potentially be seen.

Level of vulnerability of the study population to mortality due to skin barrier compromise may have been too low overall to measure population-level impact. As prematurity and LBW becomes more extreme, the skin barrier is increasingly immature, easily damaged, and functionally compromised [6–10]. Further, this is associated with increased risk of mortality, due in part to accelerated losses of water and heat through the skin and compromised skin barrier protection again the entry of pathogens through the skin [2,3,23,24]. The infants most vulnerable to skin barrier compromise—those who were VLBW and also likely to be very preterm—constituted only 2.2% of the study population and 5.2% of deaths in our study—too few to drive measurable change in population-level mortality on their own. The next most vulnerable category of LBW infants weighing 1,501 to 2,500 g, while a sizeable 38.6% ($n$ = 10,269) of our study population, is typically comprised of a significant proportion of term intrauterine growth restricted infants in the South Asian context [25,26] whose skin barrier function is relatively unimpaired [27–29]. This group had 32.1% of the deaths in the study population, reflecting their relatively low vulnerability compared to the VLBW infants (Table 2). The current evidence for impact of emollient therapy on newborn outcomes was generated in hospitalized very preterm infants based on established evidence of compromised skin barrier function. Subgroup analysis that focused on the most vulnerable VLBW infants ≤1,500 g found a

significant 52% reduction in mortality in the SSO arm. Adherence to recommended treatment of exclusive SSO application in this subgroup was particularly low (23.5% overall); however, 76.8% of VLBW intervention infants received SSO at least once versus 1.8% of infants in the comparison group. This reflects the possibility of enhanced benefit from SSO therapy, even with partial adherence, in infants with the highest level of skin barrier compromise and vulnerability.

These findings highlight that the design of studies on emollient therapy must carefully consider 3 important factors: efficacy, adherence, and vulnerability. The findings from our study when interpreted in the light of previous efficacy studies in hospitalized very preterm infants suggest that future assessments of impact on NMR must ensure that the emollient under evaluation is applied with high adherence in highly vulnerable infants.

Our study had several limitations and important considerations in interpreting the results. Unlike previous hospital-based efficacy studies, intervention workers promoted but did not apply the SSO themselves, and measurement of emollient use relied on retrospective interviews from caregivers. Our results therefore represent the expected results of large-scale promotion of SSO therapy but should not be interpreted as results of direct application of SSO therapy. The intervention design prevented us from masking allocation to study workers or participants. Thus, despite our best efforts, care provided to or evaluation of intervention infants may have differed from comparison infants in ways that could not be adequately accounted for. However, we took multiple measures throughout the trial to minimize bias and Hawthorne effect and ensure that any effects measured were solely attributable to the intervention. In hindsight, the choice of post-24-hour NMR as one of the primary outcomes of the study was a poor one. Although introduced with the intent to improve the power of the study, the assumption of delay in accrual of benefit was not observed (e.g., Fig 3B). The decision to choose 24 hours as the cutoff point was convenient but arbitrary and not recommended for future studies. Certain assumptions in the sample size calculation also limited the precision and power of the study. We had assumed cluster sizes to be equal, while in reality, the cluster size ranged from 15 to 492 live-born infants. The assumed value of 0.10 for the between-cluster coefficient of variation was one-fifth of the observed value of 0.5. Premature termination of the study further reduced the available sample size.

Subgroup and per-protocol analyses may be subject to bias. It is challenging to identify and obtain permission to weigh infants who die during the neonatal period, especially those who die early and in the community. Thus, we were missing the birth weight of 29.3% of infants who died. While there is no apparent reason for infants with missing birth weights to be differentially distributed by study arm, the subgroup analysis on VLBW infants only included 2.2% of all the infants enrolled in the study, and, therefore, sampling bias cannot be ruled out. It was not feasible to reliably capture infant weight at the time of birth, and, thus, we used weight at the first postnatal visit as a proxy for birth weight. There was on average about a 2-day period from birth to weighing, and the weights that we recorded may be slightly lower than actual birth weight [30]. Thus, subgroup analyses ($\leq$2,500 g, $\leq$1,500 g) may include small percentages of infants whose birth weights were above those cutoffs at the time of birth, although this would be expected to have minimal to no impact on the findings.

Per-protocol analysis better approximates actual emollient application rather than its promotion and included about one-third of all infants enrolled. Given the scale of the study, measurement of emollient use was based on caregiver responses and not the actual observation of oil application, and, therefore, may be subject to respondent bias. It is also possible that families that not only accepted SSO but also adhered to its exclusive use may be more likely to adhere to general recommendations on newborn care by formal health providers and health workers, and, thus, the per-protocol analysis could be comparing 2 populations with different

motivations and health attitudes, thus potentially overestimating the actual impact of the intervention.

The mechanism of impact of SSO therapy on newborn survival is biologically plausible and may be multifactorial, but requires further research [12,31,32]. The intervention was promoted as a package of SSO plus gentle massage, and, thus, we cannot disentangle the separate impacts of the oil from the way in which it was applied. However, it is likely that they interact and that both are important in intervention promotion [31]. Evidence furthermore suggests that emollient therapy has benefits for very preterm infants beyond that provided by massage alone [33–35]. Lipids in SSO may have been absorbed into and through the skin and exerted local and systemic metabolic effects. The effects may include reduction in risk for skin injury and improvement in skin barrier function, reduction in water and energy loss through the skin, impedance of transcutaneous invasion of pathogens, amelioration of essential fatty acid deficiency, provision of calories from triglyceride absorption, and/or improvement in immune function, particularly innate antimicrobial barrier defense [31,36–38]. Linoleic acid—the primary fatty acid component in SSO—binds specifically to receptors in keratinocytes that mediate skin development, and, thus, accelerates this process [39–41]. While products with skin barrier enhancing properties have been identified, it is likely that efficacy of topical products for newborn survival, health, and well-being could be further improved, for example, through formulations that lower pH and provide an optimal balance of lipids [11,12,42].

Overall, our study did not find benefit in universal promotion of SSO therapy for mortality reduction among all newborn infants in community settings, and we therefore do not currently recommend it as a public health strategy for NMR reduction. However, due to the lower than expected adherence to exclusive SSO application, potential benefit of exclusive application of SSO over mustard oil cannot be ruled out and needs to be systematically investigated. The subanalysis showing a 52% reduction in mortality in VLBW infants is consistent with prior hospital-based efficacy studies with very preterm infants (<33 weeks gestational age). The comparative lack of effect among the larger subgroup of LBW infants (1,501 to 2,500 g) suggests that in community settings, SSO therapy may only be effective for mortality reduction among newborn infants with highly compromised skin barrier function. In order to advance the science on newborn emollient therapy and its applicability along the continuum of care, it will be important to systematically disentangle the independent effects of emollients and skin care practices, including massage in infants with highly compromised skin barrier function, of varying gestational ages and under different environmental conditions.

## Supporting information

**S1 CONSORT Checklist. The trial checklist for the CONSORT Extension for Cluster Trials 2012 (based on http://www.consort-statement.org/extensions?ContentWidgetId=554).** CONSORT, Consolidated Standards of Reporting Trials.
(DOCX)

**S1 Analysis Plan. The original analysis plan as defined in the trial protocol along with any modifications.**
(DOCX)

## Acknowledgments

We thank the community of Shivgarh for their participation and enthusiastic involvement in the study. We thank the National Health Mission, Uttar Pradesh, in particular, the late Dr. Anil Kumar Verma for his cooperation and support. We would like to thank Dr. Cyril

Engmann and Dr. Jana Patterson for their support. We appreciate the extraordinary support and guidance from Dr. Rajiv Bahl and Dr. Jose Martines from WHO throughout the design and conduct of the study. We are immensely grateful to the late Prof. Maharaj Kishan Bhan as the chair of the TAG for critical inputs in the design and conduct of the trial. We would like to express our sincere appreciation to the DSMB chaired by Dr. Vinod Paul, with participation from Drs. Nita Bhandari, C. M. Pandey, and Sunil Sazawal, for monitoring the conduct and safety of the trial. We are immensely grateful to Dr. Sanjeev Agarwal for coordinating the supply of cold-pressed SSO. We thank Prof. Jai Vir Singh and Prof. Monica Agarwal for their valuable support in the coding of verbal autopsies. We thank Sachiyo Yoshida at WHO for her support with data management. We thank Raja Rakesh Pratap Singh of Shivgarh for his unflinching support. We are also grateful to Dr. Ramesh C. Ahuja and Dr. Girdhar G. Agarwal for their critical inputs and mentorship. We thank Hitesh Mahajan, Ranjit Kumar, and Col Fasihuddin Ahmed for their significant contributions and Sharat Pradhan for his invaluable support. Hina Mehrotra, Pawankumar Patil, Arti Sahu, Pramod Singh, Vivek Singh, Ranjana Yadav, and Sharad Yadav, as part of the Shivgarh Emollient Research Group, made significant contributions to the project in various capacities.

## Author Contributions

**Conceptualization:** Aarti Kumar, Vishwajeet Kumar.

**Data curation:** Shambhavi Mishra, Sana Ashraf.

**Formal analysis:** Shambhavi Mishra, Sana Ashraf, Peiyi Kan.

**Funding acquisition:** Vishwajeet Kumar.

**Investigation:** Aarti Kumar, Vishwajeet Kumar.

**Methodology:** Aarti Kumar, Shambhavi Singh, Amit Kumar Ghosh, Alok Kumar, Raghav Krishna, Vishwajeet Kumar.

**Resources:** David K. Stevenson, Lu Tian, Peter M. Elias, Gary L. Darmstadt.

**Supervision:** Aarti Kumar, Vishwajeet Kumar.

**Validation:** David K. Stevenson, Lu Tian, Peter M. Elias, Gary L. Darmstadt.

**Writing – original draft:** Aarti Kumar, Gary L. Darmstadt, Vishwajeet Kumar.

**Writing – review & editing:** Aarti Kumar, Shambhavi Mishra, Shambhavi Singh, Sana Ashraf, Peiyi Kan, Amit Kumar Ghosh, Alok Kumar, Raghav Krishna, David K. Stevenson, Lu Tian, Peter M. Elias, Gary L. Darmstadt, Vishwajeet Kumar.

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
