## [Editor Report · Decision Letter 0]

14 Oct 2020

Dear Dr Darmstadt , 

Thank you for submitting your manuscript entitled "Impact of emollient therapy with sunflower seed oil on survival of newborn infants in Uttar Pradesh, India: a community-based, cluster randomized, open-label controlled trial" for consideration by PLOS Medicine.

Your manuscript has now been evaluated by the PLOS Medicine editorial staff and the Special Issue guest editors, and I am writing to let you know that we would like to send your submission out for external peer review in the first instance.

Kind regards,

Artur Arikainen,

Associate Editor

PLOS Medicine

---

## [Decision Letter · Decision Letter 1]

19 Nov 2020

Dear Dr. Darmstadt ,

Thank you very much for submitting your manuscript "Impact of emollient therapy with sunflower seed oil on survival of newborn infants in Uttar Pradesh, India: a community-based, cluster randomized, open-label controlled trial" (PMEDICINE-D-20-04871R1) for consideration at PLOS Medicine. 

[LINK]

In light of these reviews, I am afraid that we will not be able to accept the manuscript for publication in the journal in its current form, but we would like to consider a revised version that addresses the reviewers' and editors' comments. Obviously we cannot make any decision about publication until we have seen the revised manuscript and your response, and we plan to seek re-review by one or more of the reviewers. 

We expect to receive your revised manuscript by Dec 10 2020 11:59PM. Please email us (plosmedicine@plos.org) if you have any questions or concerns.

We look forward to receiving your revised manuscript. 

Sincerely,

Artur Arikainen, 

Associate Editor 

PLOS Medicine

plosmedicine.org

1. Please address the reviewers’ comments below.

2. Financial Disclosure: Please add the following, if true (otherwise please clarify): “The funders had no role in study design, data collection and analysis, decision to publish, or preparation of the manuscript.”

3. Data Availability: We recommend that you deposit your data to a more permanent repository than Dropbox. Some options (most are free of charge) can be found here: https://journals.plos.org/plosmedicine/s/recommended-repositories. 

4. Abstract:

a. Please include trial recruitment dates.

b. Around line 50: Please describe the primary and the main secondary objectives.

c. Please provide summary participant demographics, eg. sex.

d. Please include a description of any adverse events, and whether they were attributed to the study.

e. In the last sentence of the Abstract Methods and Findings section, please describe the main limitation(s) of the study's methodology.

f. Line 65: Please start with “In this trial, we observed that…”

6. Please use the "Vancouver" style for reference formatting, and see our website for other reference guidelines https://journals.plos.org/plosmedicine/s/submission-guidelines#loc-references. Citations should be in square brackets and not superscript.

7. Line 173: Please clarify whether written informed consent for trial participation was obtained from parents/guardians.

8. Lines 301-307: Please remove funding details here – this should be included on the online submission form instead.

9. Line 385: Please clarify: “This is the first community-based trial, to our knowledge, to report…”

10. Lines 537-568: Please delete these sections here - they should be included on the online submission form instead.

11. Figure 2 legend: Please rename “Participant flow diagram”.

12. Please complete the CONSORT checklist, include it as a Supporting Information file, and ensure that all components of CONSORT are present in the manuscript. When completing the checklist, please use section and paragraph numbers, rather than page numbers.

13. Please include the trial protocol as a Supporting Information file, and add a citation to it in the Methods.

---

Comments from the reviewers:

Reviewer #1: Please see the attached word document.

Reviewer #2: Your article is very well written. It is a valuable work and it will definitely be very useful for your community. 

It has no methodological, scientific or executive problems. I have no comments in these areas.

There is only one problem : this intervention is known to some extent and all researchers recognize its effectiveness. What kind of innovation or difference is in your activity and research that distinguishes it from others? Please include the importance of study and its necessity in the introduction section.

 It may still be a questionable problem or have a particular importance to your community.

Best regards,

Reviewer #3: This is a well-conducted Cluster RCT on the impact of emollient therapy with sunflower seed oil on survival of newborn infants in Uttar Pradesh, India. The study design, sample size, statistical methods and analyses, and presentation (tables and figures) and interpretation of the results are mostly adequate and of a good standard. However, there are still a few issues needing attention.

1) Need to make it absolutely clear the study uses co-primary outcomes including both NMR and post-24-hour NMR.

2) Sample size section on page 6. Need a bit more details and clarification how the sample size was calculated as need to satisfy both co-primary outcomes so may need to calculate separately and then combine and choose one sample size for both. Need justification/reference for all the parameters used for the sample size calculation. Why ICC is 0.1? Why assumed a mean NMR of 45 per 1000? and etc.

3) Primary analysis on page 10-11. It's a bit confusing as both logistical regression and survival analysis (proportional hazard model) were used. There can only be one primary analysis, and other one (survival) need to be toned down as sensitivity check. The results are both negative, otherwise (one negative, one positive) will be confusing and inconsistent.

4) Table 1. Some of the continuous variables are not normally distribution (skewed) such as cluster size and age at measurement of first visit weight, therefore need to present as median and IQR rather than mean and SD.

[LINK]

---

## [Decision Letter · Decision Letter 2]

8 Mar 2021

Dear Dr. Kumar,

Thank you very much for re-submitting your revised manuscript "Impact of emollient therapy with sunflower seed oil on survival of newborn infants in Uttar Pradesh, India: a community-based, cluster randomized, open-label controlled trial" (PMEDICINE-D-20-04871R2) for consideration in PLOS Medicine’s Special Issue on Global Child Health.

Your paper was evaluated and discussed among all the editors here. It was also discussed with an academic editor with relevant expertise, and sent to two reviewers, including a statistical reviewer. The reviews are appended at the bottom of this email and any accompanying reviewer attachments can be seen via the link below:

[LINK]

Thank you for addressing the original set of comments from the reviewers. There are a few remaining points from the reviewers that need to be addressed, we would like to consider a revised version that addresses the reviewers' and editors' comments. In particular, please address Reviewer 3's request regarding correction for cluster size and age.

Obviously we cannot make any decision about publication until we have seen the revised manuscript and your response, and we may seek re-review by one or more of the reviewers. 

We expect to receive your revised manuscript by Mar 15 2021 11:59PM. Please email us (plosmedicine@plos.org) if you have any questions or concerns.

We look forward to receiving your revised manuscript. 

Sincerely,

Caitlin Moyer, Ph.D.

Associate Editor 

PLOS Medicine

plosmedicine.org

1. Abstract: Methods and findings: Please note which variables were adjusted for in reporting the adjusted odds ratios for the primary outcome.

2. Throughout: Please use consistent terms for children (“newborn infant” vs. “babies” etc.)

3. Author summary: Why was this study done?: We suggest combining/shortening some of the bullet points in this section- for example, points 2 and 3 could be combined into a single point briefly describing the background/rationale for the study, and the final point could more succinctly describe the study objective/hypothesis.

4. Author summary: What did the researchers do and find? Again, for this section, we suggest fewer- potentially three bullet points: 1) What was done; 2) primary findings; 3) secondary findings.

5. Author summary: What do these findings mean? We suggest combining the first three points into 1-2 points as follows:

-Our findings do not support recommendations for universal promotion of SSO therapy under typical implementation conditions for mortality reduction. However, the low levels of adherence to recommended treatment and a significant mortality reduction by per protocol analysis that we observed suggests that a potential benefit of SSO over mustard oil cannot be ruled out and needs systematic investigation.

-Results from our sub-group analysis on the highest-risk category of very low birth weight infants are consistent with previous hospital-based studies and potentially extend the applicability of emollient therapy in this sub-group to the facility community Continuum.

6. Methods: Please add the following statement, or similar, to the Methods: "This study is reported as per the Consolidated Standards of Reporting Trials (CONSORT) guideline (S1 Checklist)."

7. Methods: On page 15 of the Methods, a predetermined analysis plan is mentioned. Please include your analysis plan as a supporting information file, noting any instances where the analyses conducted were different from those outlined.

8. Results: Page 19: Frequency of massage: Please indicate if the times reported are in times/day or other unit (“...intervention infants were massaged (regardless of oil used) for a mean of 2.68 (95% CI 2.66– 2.70) times [median and IQR 3±1]...”)

9. Results: Page 20: Intention to treat analysis of NMR: Please provide units for the NMR values reported here (“...did not find a significant difference in mortality among newborn infants in the intervention (NMR 53.7 ± 2.3)...”

10. Discussion: Page 25: Please revise and clarify this sentence if possible, as the meaning is slightly unclear: “In retrospect, more appropriate study designs than that used here may have been an efficacy trial in all newborn infants which ensured that all infants received the therapy as intended and examined the population-level mortality effect, or a population-based effectiveness trial targeted to highly vulnerable VLBW infants – a group in whom the treatment has shown efficacy under hospital conditions – but under conditions of public health delivery with an eye toward scalability by the health system or potential through markets if the intervention was found promising.”

11. Table 2: Please also provide the unadjusted results (this may be in a separate/supporting information table if preferred).

12. Figure 3: Please note in the legend that the graph in panel B has a different y axis scale than the other two.

13. CONSORT Checklist: Thank you for including the CONSORT checklist, please revise the checklist using section/paragraph numbers to refer to locations in the text, rather than page numbers.

Comments from the reviewers:

Reviewer #1: Please see the attached file.

Reviewer #3: Many thanks authors for their great effort to improve the manuscript. The authors have addressed my comments very well. I am mostly statisfied with the response and revision. Only one very minor point remaining. On point 4 using median and IQR for non-normal variables (Table 1), it should be presended, for example, 80 (60-113) rather than 80±53 (i don't know exactly the IQR so just give an example). IQR is a range so need to present as medain (xx-yy). Need to correct for both cluster size and age. It's a very minor point. Depending on the editor, as long as it's corrected may not need to come back to me again.

[LINK]

---

## [Decision Letter · Decision Letter 3]

27 Apr 2021

Dear Dr. Kumar,

Thank you very much for re-submitting your manuscript "Impact of emollient therapy with sunflower seed oil on survival of newborn infants in Uttar Pradesh, India: a community-based, cluster randomized, open-label controlled trial" (PMEDICINE-D-20-04871R3) for consideration in PLOS Medicine’s Special Issue: Global Child Health: From Birth to Adolescence and Beyond.

I have discussed the paper with my colleagues and the academic editor and it was also seen again by two reviewers. I am pleased to say that provided the remaining editorial and production issues are dealt with we are planning to accept the paper for publication in the journal.

[LINK]

We look forward to receiving the revised manuscript by May 04 2021 11:59PM.   

Sincerely,

Caitlin Moyer, Ph.D.

Associate Editor 

PLOS Medicine

plosmedicine.org

Requests from Editors:

1. Title: Please revise the title slightly, we suggest: “Effect of sunflower seed oil emollient therapy on newborn infant survival in Uttar Pradesh, India: A community-based, cluster randomized, open-label controlled trial” or similar.

2. Data availability statement: Thank you for providing a link to access the dataset. It seems as if download of the data file is restricted. Please indicate whether this restriction will be removed. If data are freely available upon request, please note this and state the owner of the data set and contact information for data requests (web or email address). Note that a study author cannot be the contact person for the data.

3. Abstract: Methods and Findings: Please change to make it completely clear that the numbers here refer to clusters and not infants: “...recommended in intervention clusters (n=138 clusters); infants in comparison clusters (n=138 clusters)...”

4. Abstract: Methods and Findings: Please indicate whether you are reporting mean +/- standard deviation: (52.2% male, mean weight 2575.0 grams ± 521.0).

5. Abstract: Methods and Findings: We suggest revising this sentence to: “...differences between the intervention and comparison groups in other pre-specified subgroup comparisons by low birth weight, birthplace, wealth and emollient use did not reach statistical significance.” or similar to clarify.

6. Method: Page 14: Data Management section: Please clarify what is meant by “observed heaping” unless this is a common term.

7. Methods: Page 14: Statistical analysis section: Thank you for including the supporting information file titled: “Prespecified analysis plan and modifications”. Please refer to this document in the text of the Methods here, for example as S1_Analysis Plan.

8. Results: Page 18:Please indicate the numbers in parentheses here are the range of days for the first visit: “At the first postnatal (enrolment) visit that occurred on day 2 (1-3) for…”

9. Discussion: Please ensure the Discussion is organized as follows: a short, clear summary of the article's findings; what the study adds to existing research and where and why the results may differ from previous research; strengths and limitations of the study; implications and next steps for research, clinical practice, and/or public policy; one-paragraph conclusion. Specifically, on page 26-27, please consider whether the discussion of SSO therapy mechanisms could be discussed earlier in the section.

10. References: Please check formatting of references 8, 9, 10 (punctuation after journal name). Please provide an update on reference 31- listed as “submitted”- please note that papers cannot be listed in the reference list until they have been accepted for publication or are publicly available on a preprint archive. Alternatively please provide a different appropriate reference. 

11. Table 1: Please define abbreviations “IQR” and “SD” in the legend.

12. Table 2: The footnote symbol for part C (exclusive mustard oil vs exclusive SSO) does not appear to be defined in the legend. Please check that the notes included in the legend match those shown in the table.

13. Figure 3 and 4: Please note the HR are presented with 95% CIs

14. Checklist/Supporting information: The CONSORT checklist is not included the set of files with the manuscript. Please include the revised version of the checklist (with section/paragraphs rather than page numbers to identify locations within the text).

Comments from Reviewers:

Reviewer #1: The authors have addressed my comments in a very thorough fashion.

Reviewer #3: Many thanks authors for the improvement. I am satisfied with the response and revision. No further issues needing attention.

[LINK]

---

## [Editor Report · Decision Letter 4]

1 Jun 2021

Dear Dr Kumar, 

On behalf of my colleagues and the Academic Editor, Quique Bassat, I am pleased to inform you that we have agreed to publish your manuscript "Effect of sunflower seed oil emollient therapy on newborn infant survival in Uttar Pradesh, India: a community-based, cluster randomized, open-label controlled trial" (PMEDICINE-D-20-04871R4) in PLOS Medicine’s Special Issue: Global Child Health: From Birth to Adolescence and Beyond.

In addition, please address the remaining editorial requests below:

1.Title: Please capitalize the first word of the subtitle: “Effect of sunflower seed oil emollient therapy on newborn infant survival in Uttar Pradesh, India: A community-based, cluster randomized, open-label controlled trial”

2. Page 29: Please remove the “Data Availability” section from the main text, as the data availability statement should be completely and accurately entered with the manuscript submission metadata, and this statement will be included with the publication.

3. References: Please provide updated information for reference 30. Please note “Forthcoming” rather than “in press” for reference 33.

4. CONSORT checklist: Please remove all references to page numbers throughout. Please use only sections/paragraph numbers to refer to locations within the text. For “Funding” please note “Financial Disclosure section” or similar.

PRESS

Sincerely, 

Caitlin Moyer, Ph.D. 

Associate Editor 

PLOS Medicine